# Assessing the quality of bee honey on the basis of melissopalynology as well as chemical analysis

Zahra Shakoori[1], Ahmadreza Mehrabian[1]*, Dariush Minai[1], Farid Salmanpour[2], Farzaneh Khajoei Nasab[1]

1 Department of Plant Science and Biotechnology, Faculty of Science and Biotechnology, Shahid Beheshti University, Tehran, Iran, 2 Department of Biodiversity and Ecosystem Management, Research Institute of Environmental Sciences, Shahid Beheshti University, Tehran, Iran

* a_mehrabian@sbu.ac.ir

**Data Availability Statement:** All relevant data are available on Figshare: https://figshare.com/s/949e02f3163f39991d4e.

## Abstract

Melissopalynological and chemical analysis of honey provide us useful and valuable information about the botanical and geographical origin of honey. The data in question is very important for authentication as well as for testing the quality of honey, so this is considered the main method in honey regulation here, we have used chemical analysis and melissopalynology to evaluate different honey samples from two main Iranian hubs of honey. Sampling was carried out on two important poles in Iran's honey production, the central Alborz region, and the mountainous Zagros ecosystems in the years 2020 to 2021. Therefore, 52 samples from Alborz (Northern Iran), as well as 42 samples from the Zagrosian ecosystems (western Iran) belonging to different ecological habitats, were collected. In addition, samples were taken at 7 altitudes from 0 to 3500 m a.s.l. Furthermore, in this study, various chemical analyses such as the effect of antioxidant activity, the amount of total phenolic content, pH, and moisture content of honey samples were evaluated. Our results showed that all honey samples were classified as polyfloral honey. Based on our findings, 57 honey samples (61%) contained the standard amount of pollen. A total of 42 plant families and 55 genera were identified in the studied samples, with the highest presence of Asteraceae, Fabaceae, Rosaceae, Apocynaceae, and Apiaceae. Finally, an antioxidant activity ratio of 19% to 98%, total phenolic content from 0.08 to 0.51 ppm, pH from 1.90 to 5.21, and moisture content from 13% to 18.40%.

## Introduction

Honey is a precious natural product that is processed by bees from the nectar of flowers and stored in the beehives. Honey production in an area depends mainly on the vegetation and climate of the region, as the flowering season and nectar production can be different for a species in different regions [1]. With the growth of the worldwide honey market, the identification of the botanical origin as well as its authenticity has received much attention in the international

**Funding:** The authors received no specific funding for this work.

**Competing interests:** The authors declare that there are no competing interests.

society [2]. Counterfeit honey and poor-quality honey are major problems in the global honey trade. Honey, which consists of many stable elements as well as pollen grains, is collected by bees along with nectar. Melissopalynology as an effective method, as well as key criteria, has been widely used to determine the purity, geographical, and botanical origin of natural honey [3, 4]. Louveaux 1970, 1978 was the first to establish this method [5, 6]. However, several laboratory procedures have been developed to study of pollen grains in honey latter [7], melissopalynology studies evaluated pollen grains in honey [8, 6, 9, 10]. Pollen analysis provides valuable data on the botanical and geographical origins of honey and is a key method for honey authentication [11]. The botanical and geographical origins of honey is important to the consumers and influences the product quality and physical, chemical, organoleptic, and bioactive properties of honey, which is considered as a valuable indicator of honey quality and authenticity [12–15]. Furthermore, the quality of honey is largely dependent on the nectariferous plant species that honey bees use in their diet. Accordingly, different melliferous plant species have different characteristics and applications (eg, drugs, nutrients, etc.) [16, 17].

Iran was ranked 9th in the world for honey production from 1993 to 2018 [18]. High climate variability, topographic complexity [19, 20], the geographical isolation of the highlands of Iran [21] as well as soil diversity [22] established different phytogeographical units in Iran [23, 24], making it the center of a endemism for plant taxa in Southwest Asia [25, 26], the center of endemism of Irano-Turanian region [23, 27, 28], and an important center of plant diversity in the world [29–32]. These ecological conditions have led to the formation of valuable beekeeping environments in the country, making honey production in Iran very diverse.

The Alborz and Zagros Mountains in Iran have created transition zones that have lead to a high diversity and endemism in this region [25, 33]. Influenced by the high-altitude diversity in this area, a variety of vegetation has been formed [34, 35], creating suitable habitats for beekeeping and honey production [36, 37]. Today, several researchers in Iran have used melissopalynology research methods, including melissopalynology study of some honey in Tehran Province [36] melissopalynology in Karaj, northern Iran [38], melissopalynology of some honey from Khorasan, N.E Iran [39], evaluation of honey pollen grains in Isfahan Province, C. Iran [40], the melissopalynology of honey in some areas of Mazandaran Province [37], melissopalynology of honey in Arasbaran, northwestern Iran [41], and origin evaluation of some Iranian honeys [42].

In addition to nutrients, honey also contains bioactive substances that can have a positive effect on the human body [43]. The healthful properties of honey are due to the presence of total phenolic content, including flavonoids, phenolic acids and their esters, as well as organic acids, free amino acids, vitamins, carotenoid derivatives, enzymes and biological elements [44]. These compounds have antioxidant activity properties [45]. Oxidation processes can be dealt with using antioxidant activity agents. Antioxidant activities can prevent many diseases or even reduce their symptoms [45]. Honey can be a complementary source of antioxidant activities in a balanced diet [46]. The antioxidant activity of honey is due to the presence of total phenolic content [47, 48]. The moisture content of bee honey is very important for its stability against fermentation and granulation. Low humidity protects honey from microbial activity and thus can be stored longer [49, 50].

The aim of the study was to evaluate different honey samples from different ecological regions of Iran based on chemical and melissopalynological analyses. In addition, the study tries to analyze the mentioned data in relation to geographical factors. Therefore, conducting a comparative study in different regions while comparing honey produced analyzes the relationship between pollen factors and some important chemical factors of honey with ecological factors.

## Materials and methods

### Study area

Iran is a prominent area of the Iranian Plateau in southwestern Asia situated between 24° to 40° N latitude and 44° to 64° E longitude, as well as a limited area of the orogenic belt (Zagros, Alborz, and other mountain chains. It covers the Asian block along with Arabian-African unity [51]. The main geomorphological areas of Iran, include Alborz, Zagros, Kopet Dagh, as well as several interior mountain ranges.

The Zagros Mountains extend from the northwest to the southeast. The Alborz Mountains are located at 36° N latitude from the Caspian Sea with an elevation of 5671 meters [52]. In addition to plant species, Alborz high forests (Hyrkanian forests) host valuable animal species such as Iranian leopards [53]. Most of our honey samples were collected from the areas of Alborz and Zagros, which have the highest honey production in Iran, including more than half of Iran's provinces, and most of the collected samples were close to forests and protected areas (Fig 1).

### Sample collection

This study is based on a direct collection of honey samples from beehives in the period from May to October 2020–2021. We selected sampling sites covering different geographical and botanical areas [54]. Using a data collection form, the geographical coordinates of the hives were recorded by a Garmin Map 64s GPS device, height, the type of honey and the claims of beekeepers. Based on the diversity of ecological conditions, bee colonies were identified and sampled from the hive. From each colony, one sample of 300 g of unrefined and non-clay honey was prepared for pollen extraction and chemical analysis. Samples were placed in a sterile sealed container and kept in a cool place until they reached the laboratory [55].

### Laboratory studies

**Material or chemical and reagents.** All the reagents for biochemical studies were obtained from Merck Chemical Company in Germany. 2,2-Diphenyl-1-picrylhydrazyl (DPPH) was obtained from Sigma American Chemical Company.

**Instrumentations.** Hot plate (model HP100 made by Mtops company in South Korea). 30 mL conical centrifuge tubes.

Centrifuge model TDL-4 made by Jenius China.

Hitachi SU3500 scanning electron microscope (SEM) made in Japan.

Spectrophotometer JENUS V-1100 made by DLAB China.

pH meter model JENWAY 3505.

Refractometer 28–62% BRIX

*Laboratory studies were performed in two parts.* First part: Melissopalynology based on scanning electron microscopy (SEM)

Second part: Chemical analysis

**First part: Melissopalynology based on scanning electron microscopy (SEM).** Melissopalynology analyzes based on the method of Louveaux et al. 1978 has been used [5]. The melissopalynological method was proposed by the International Commission on Bee Botany (ICBB) [5, 56]. First, 10 grams of honey in about 20 mL of distilled water was dissolved on a hot plate at a temperature below 40°C. Then, the honey samples were filtrateto remove large honey wax particles, the liquid was poured into 30 mL conical centrifuge tubes. After several centrifugation times (2500–3000 rpm,) at room temperature (21 to 25°C), pollen grains were extracted from the honey. In the first round, the samples were centrifuged for 10 min

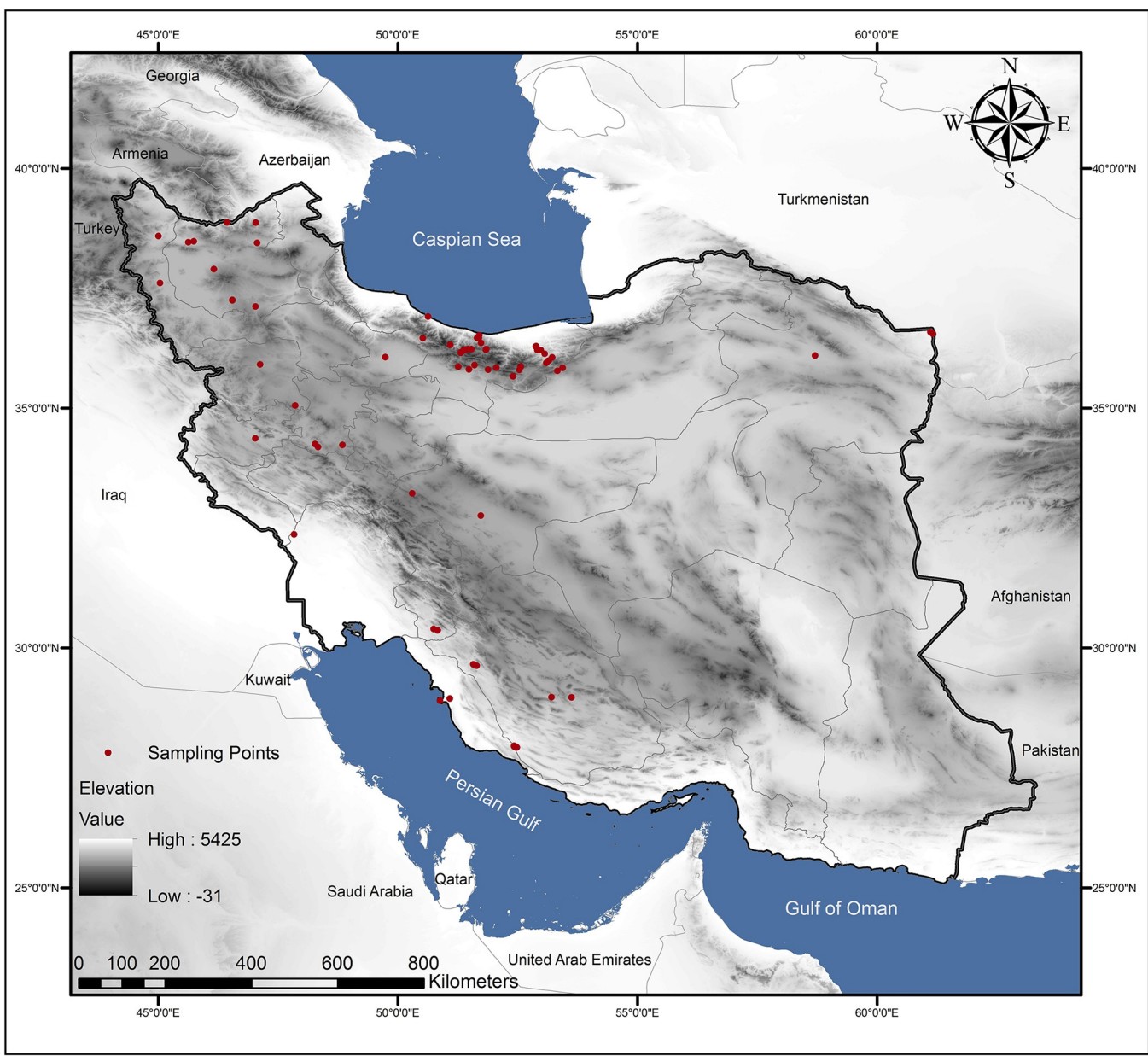

**Fig 1. Study area and sample collection points.** All the sampling locations are marked using red dots.

(approximately 2500 rpm) and the supernatant was carefully removed to avoid pollen loss [55]. The remaining precipitate was mixed again with about 10 mL of distilled water, but this time the samples were centrifuged for 5 min (2500–3000 rpm). In our study, we repeated this process 4–7 times depending on the amount of sugar in the sample to completely remove the sugar in the honey. But after the last centrifugation, we poured the last remaining liquid into the watch glass and placed it on the hot plate so that the water evaporated completely and the pollen grains remained at the bottom of the container. During this process, extreme care should be taken to avoid any contamination from external pollen from previous honey products (disposable equipment is recommended whenever possible) or pollen grains in the air (close windows and limit exposure).

We placed the prepared pollen grains on special bases and, after coating the samples with gold metal, we placed them inside the scanning electron microscope (SEM). Depending on the size of the sample, using this microscope, it is possible to shoot surfaces with magnification from 10 to 500,000 times and resolutions below 1 to 20 nanometers. By analyzing the information obtained from SEM, we identified and counted the pollen grains, and then obtained the botanical origin of the honey.

*Identification and counting.* By using SEM, pollen grains obtained from honey sediments were counted, Counting 500 to 1000 pollen grains are necessary to determine relative abundance [57]. For pollen identification we used pollen atlases and websites, the identification of pollen grains was done according to the morphology of the pollen grains. Pollen species should only be referred to as botanical species or species if they can be reliably identified at the genus or species level, which is rare [56].

**Second part: Chemical analysis.** In this part of the study, several chemical analyses of honey such as antioxidant activity, total phenolic content, pH, and moisture content were investigated. For all chemical analyses, honey was diluted 1/1 (w/v) with distilled water.

*Analysis of antioxidant activity by DPPH free radical scavenging activity.* The DPPH method was used to analyze all 94 honey samples [58].We dissolved a concentration of 0.1 mM of DPPH in methanol, and a purple solution was obtained. Then, 100 μL of DPPH was mixed with 10 μL of honey diluted and was incubated for 30 min in the dark. Absorption was recorded at 520 nm using a spectrophotometer. Ascorbic acid (20 mg/mL) was used as a positive control. The antioxidant activity was calculated from the following formula.

$$DPPH\ scavenging\ activity\ (\%) = A_{control} - A_{sample}/A_{control} \times 100$$

*Analysis of total phenolic content.* 0.1 ml of ferric acid ($FeCl_3$) was mixed with 0.1 ml of diluted honey and the absorbance was measured at 590 nm., against Phenol solution (0.09%) which was used as standard [59].

*Measuring pH and the moisture of honey.* To determine the pH, 1 g of honey was mixed with 1 mL of distilled water, and the pH of each sample was measured with a pH meter and the moisture content of pure honey was determined with a refractometer. Using the Minitab software, we performed a regression between the mean annual relative humidity of the honey collection sites and the moisture content of the honey samples using the fitted line plots statistical model.

**Data analysis.** *Principal component analysis (PCA).* Principal component analysis (PCA) was used to determine the relationship between the geographical origin of honey samples and the amount of chemical compounds they contained. The main objective of this analysis was to determine whether honey samples in a geographic area contained the same amount of chemical compounds. Chemical parameters (pH), chemical compounds (total phenolic content and antioxidant activity) and moisture content were used for PCA analysis. A variance-covariance matrix was generated for all honey samples and four chemical parameters. Data were analyzed using PAST software version 3.21. [60].

*Cluster analysis.* We used clustering to determine the similarity of the honey samples studied based on the composition of the plant families contained within them. A variance-covariance matrix was generated for all the honey samples and their botanical families. The studied samples were clustered using the Euclidean distance and UPGMA (unweighted pair group method using arithmetic average) using the PAST software.

## Results and discussion

### Pollen analysis

In this study, after analyzing pollen grains of 94 honey samples, a total of 42 plant families were identified. Among them, Asteraceae (38%), Fabaceae (13.5%), Rosaceae (6%), and Apocynaceae and Apiaceae (5%) were the largest plant families (Fig 2). Fig 3 illustrated some pollen images of the Asteraceae, Fabaceae, Rosaceae, and Apocynaceae families (Fig 3A–3D). The mentioned results show strong similarity with the following studies: Karimi in Fars Province [61] and Razzaghy in Mazandaran Province [37]. However, Asteraceae, Fabaceae, Rosaceae, Apiaceae, Brassicaceae, Papilionaceae, and Lamiaceae, respectively, showed the highest presence, in the research of Rašić [62]. Furthermore, in studies by Puusepp and Koff in Estonia [63], plant species of the Asteraceae, Fabaceae, and Rosaceae families showed the strongest presence. In addition, 55 plant genera were identified (Table 1).

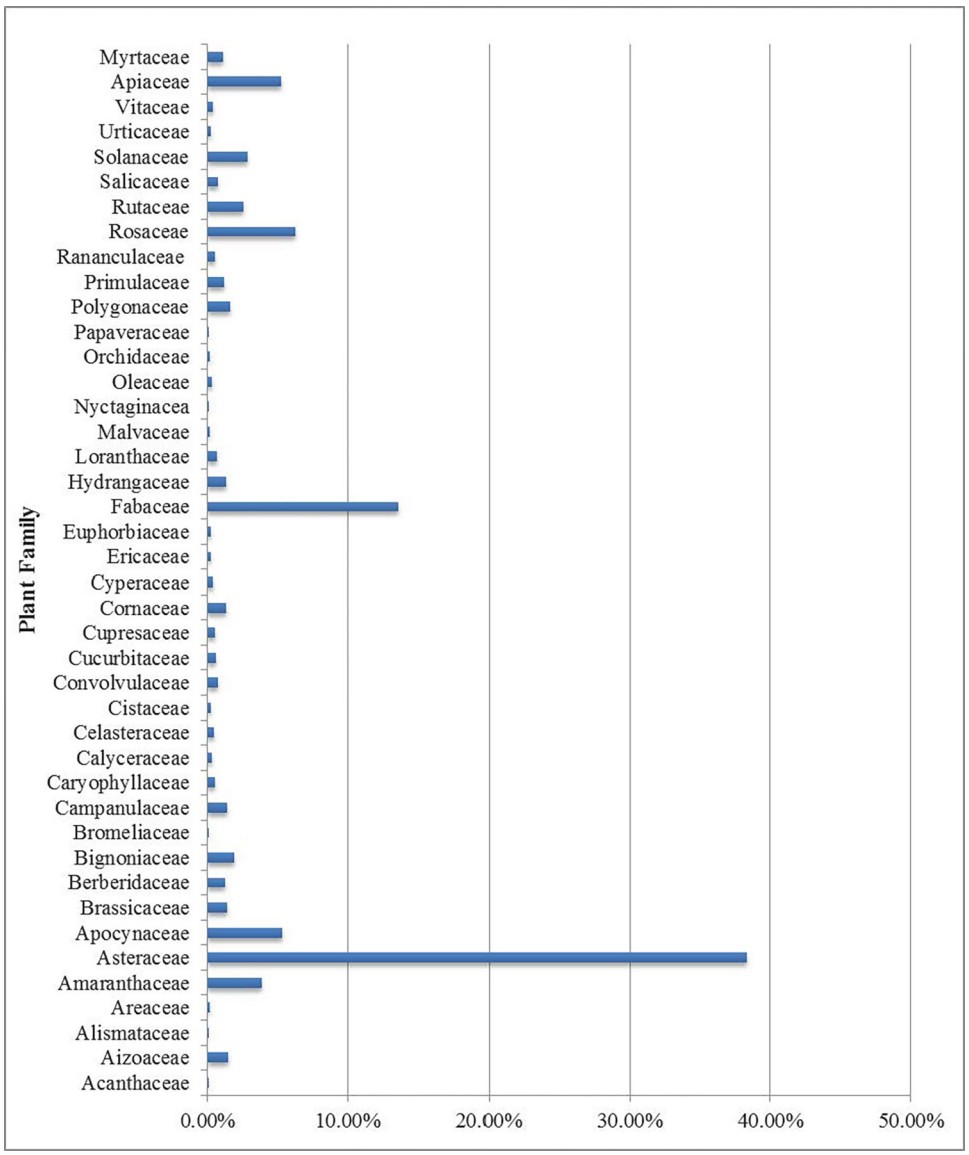

**Fig 2. Percentage of presence of each plant family that was identified through analysis of pollen of 94 honey samples.**

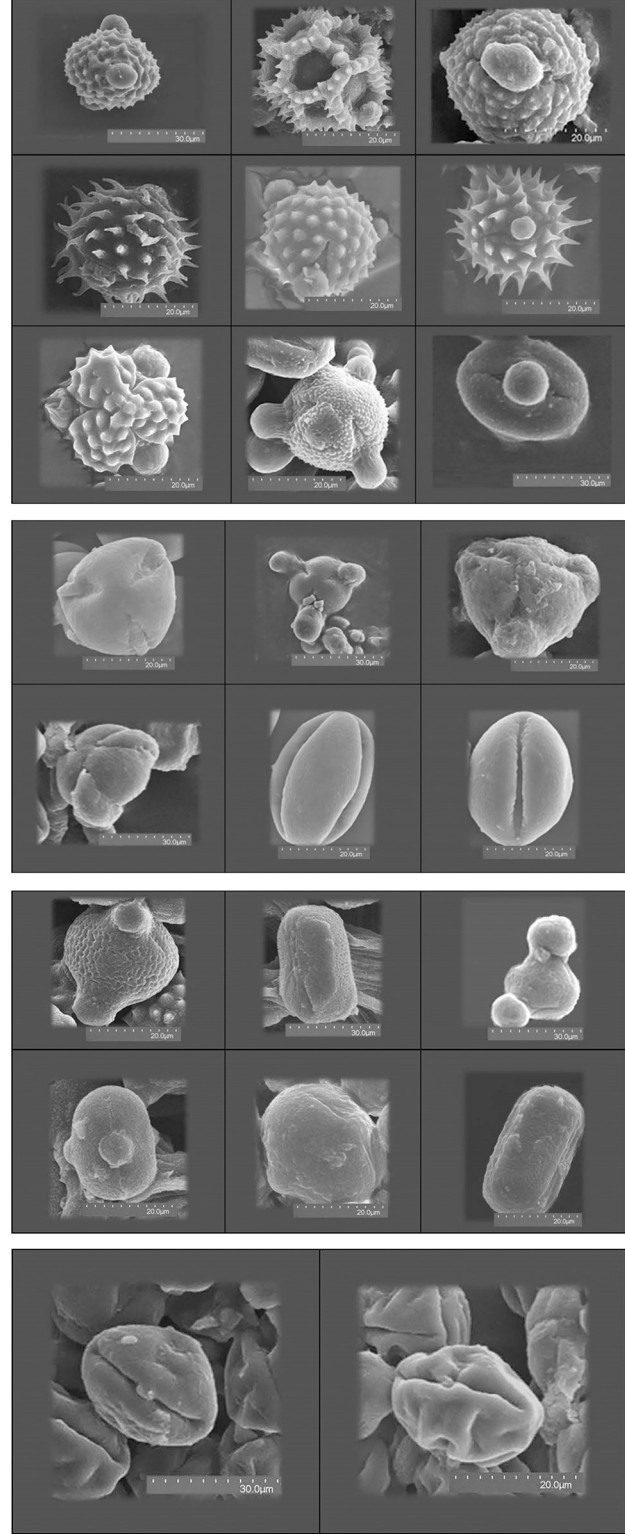

**Fig 3. A.** Some pollen images of the Asteraceae family. **B.** Some pollen images of the Rosaceae family. **C.** Some pollen images of the Fabaceae family. **D.** some pollen images of the Apocynaceae family.

**Table 1. Pollen types of 94 honey samples in this study.**

| Family | Samples Number | Pollen type |
|---|---|---|
| **Amaranthaceae** | N11, N14, N15,N21,N25, N27, N28, N30, N31 | *Amaranthus* sp. |
| | | *Atriplex* type |
| **Apocynaceae** | C2.2, F1, F4, G3, H3, H4, K3, L4, N27, N28, N31 | *Catharanthus* type |
| **Apiaceae** | N2, N3, N5, N6, N7, N10, N11, N15, N16, N30, N36 | *Apiaceae* type |
| | | *Apiaceae* type |
| | | *Torilis* sp. |
| **Acanthaceae** | N14 | *Acanthaceae* type |
| | A1.1, B4, C2.3, D2, E2, E3, F2, F4, H1, H3, H4, K4, L2, M4, N1, N2, N3, N5, N6, N7, N9, N10, N11, N13, N15, N16, N17, N19, N20, N21, N22, N23, N24, N25, N26, N28, N30, N31, N36, N37, N38, N39, N41, N42 | *Achillea* sp. |
| | | *Artemisia* sp. |
| | | *Carduus* type |
| | | *Carthamus* type |
| **Asteraceae** | | *Centaurea* type |
| | | *Cirsium arvense* |
| | | *Crepis* sp. |
| | | *Doronicum* sp. |
| | | *Helichrysum* sp. |
| | | *Hieracium* sp. |
| | | *Inula* sp. |
| | | *Senecio* sp. |
| **Bignoniaceae** | C2.2, I2 | *Bignoniaceae* type 1 |
| | | *Bignoniaceae* type 2 |
| **Brassicaceae** | F4, G3, L4, N2, N6, N33 | *Brassicaceae* type |
| **Caryophyllaceae** | N26, N31 | *Caryophyllaceae* type |
| **Cistaceae** | C2.1 | *Cistus* sp. |
| **Convolvulaceae** | N21, N29 | *Convolvulus arvensis* |
| **Cornaceae** | C1.1 | *Cornus sanguinea* |
| **Cucurbitaceae** | A1.3, G2 | *Cucurbita* sp. |
| **Cyperaceae** | N25, N30 | *Carex* sp. |
| | A1.1, D1, F1, F4, G2, H1, H3, H4, L4, M4, N1, N3, N6, N9, N14, N15, N17, N18, N19, N22, N23, N26, N30, N31, N32, N38, N39 | *Astragalus* type |
| | | *Robinia pseudacacia* |
| **Fabaceae** | | *Hedysarum* sp. |
| | | *Onobrychis cornuta* |
| | | *Vicia* sp. |
| **Hydrangeaceae** | B3, N7, N14 | *Philadelphus* sp. |
| **Myrtaceae** | N6, N33 | *Eucalyptus calmadonensis* |
| **Oleaceae** | I1, N1, N30 | *Olea europaea* |
| | A1.1, D2, F4, G3, N6, N14, N30 | *Fagopyrum* sp. |
| **Polygonaceae** | | *Polygonum* sp. |
| | | *Rumex* sp. |
| | A1.1 | *Adonis* sp. |
| **Ranunculaceae** | | *Clematis* sp. |
| | | *Ranunculus* sp. |

(*Continued*)

**Table 1.** (Continued)

| Family | Samples Number | Pollen type |
|---|---|---|
| | A1.1, B3, C2.2, D1, F4, G2, H1, H3, H4, L4, N1, N6, N10, N14, N15, N20, N27, N39, N41 | *Alchemilla* sp. |
| | | *Cotoneaster* sp. |
| | | *Sorbaria* sp. |
| **Rosaceae** | | *Prunus domestica* |
| | | *Rosa cania* |
| | | *Sorbaria* sp. |
| | | *Spiraea* sp. |
| **Rutaceae** | C2.1, E3, F4, N14 | *Rutaceae* type |
| | | *Poncirus* sp. |
| **Salicaceae** | A1.3, N29 | *Populus nigra* |
| **Solanaceae** | C2.2, E1, E3, I1, N26 | *Solanum* spp. |
| **Urticaceae** | N22, N27 | *Parietaria judaica* |
| | | *Urtica dioica* |

Our results showed that all the studied samples are classified as polyfloral honey and based on the palynological criteria, about 30% of the studied honey samples were not of good quality, the rest good quality.

Pollen grains extracted from honey samples were identified based on pollen references. A quantitative assessment of pollen grains in each honey sample was performed according to the Louveaux standard. Samples were collected at altitudes ranging from 12 to 3062 meters above sea level. Samples were taken at seven altitudes, considering the variation of vegetation type and the location of beekeepers depending on the season. More than 73% of the samples were taken at altitudes between 1000 and 3000 meters above sea level (Fig 4).

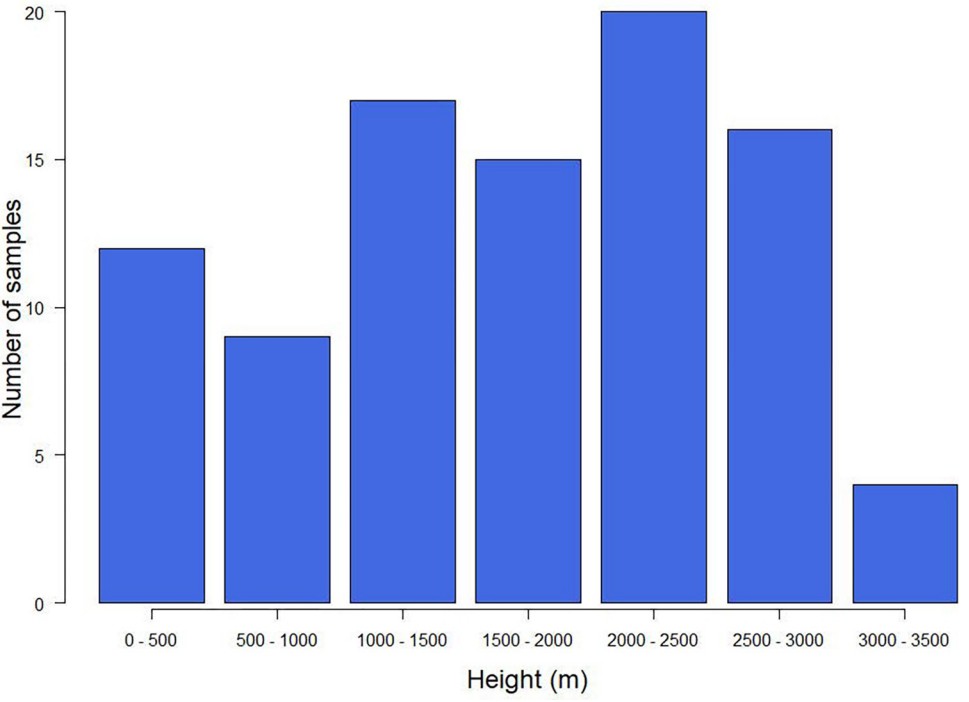

**Fig 4. Number of samplings performed at different altitude intervals.**

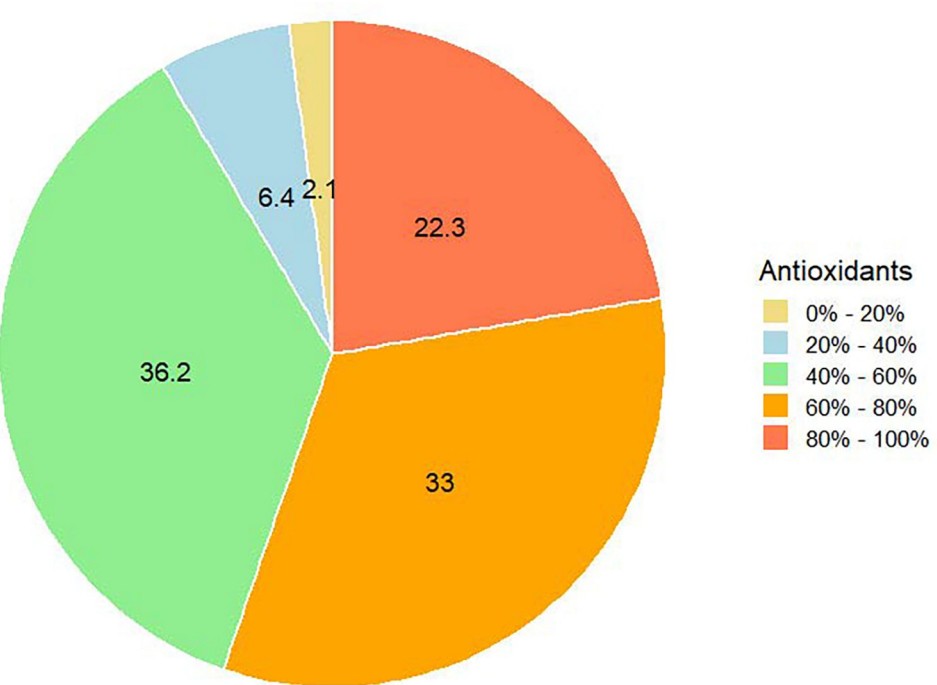

**Fig 5. The antioxidant activity abundance ratio of honey samples.**

**Physicochemical analysis of honeys.** According to the results of the chemical analysis of honey samples, the amount of antioxidant activities in the samples was as low as 19% and the highest as 98%. According to Fig 5, more than 36% of the total samples had antioxidant activities between 40% and 60%.

The chemical, physical, antioxidant and health-promoting properties of honey vary depending on the flower source, soil type, geological root, and climatic conditions CACS (2001). Ancient and modern civilizations used honey as medicine for various ailments [64]. It is also a delicious food composed mainly of various sugars and compounds such as proteins, organic acids, amino acids, nutrients, minerals, carotenes and aromatic substances. It is rich in flavonoids and phenolic acids, exhibits many biological effects and acts as natural antioxidant activity [65]. The results of our chemical analysis showed that the chemical composition of honey samples collected from different ecological regions was different from each other [66]. Chemical experiments also showed that about 3/5 of the samples showed more than 60% of the effect of antioxidant activity. We compared the results of antioxidant activities with increasing elevation, which seems to have a positive response to increasing height, and up to 3000 meters the trendline showed an upward trend, but after 3000 meters, the antioxidant activities followed the exact opposite trend. The highest efficiency of antioxidant activity was observed at altitudes between 2500 and 3000 meters, with an average of 88.71% (Fig 6). The potential variation of the antioxidant activity of honey depends on topographical and botanical differences [67, 68]. The antioxidant activity of honey is mainly due to phenolic and flavonoid compounds. Thus, a significant difference in antioxidant activity is observed between different types of honey worldwide [36, 69, 70].

The total phenolic content in the honey samples was between 0.08 and 0.59 micrograms. Differences in total phenolic content in honey samples may be due to different geographical origins or other factors such as collection season and storage conditions. An important group

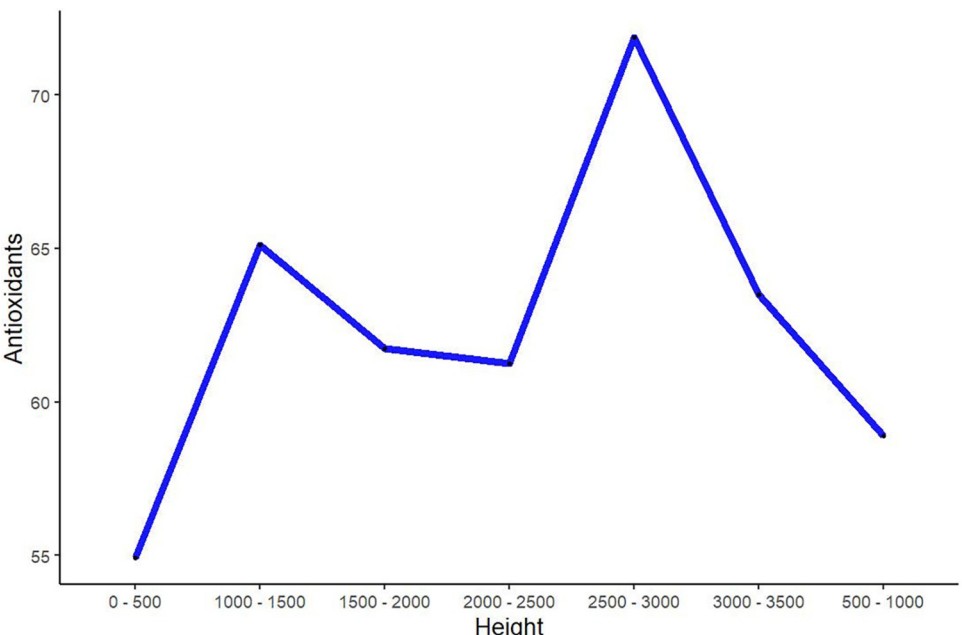

**Fig 6. Comparison of antioxidant activities with altitude.**

of compounds affecting the appearance and functional properties of honey is phenolic acid, which has nutritional value and medicinal properties [71]. The type and concentration of total phenolic content in different kinds of honey are different from each other [72]. The amount of total phenolic content is influenced by various factors such as plant type, geographical origin, and climatic conditions of the honey production site [73]. Determination of total phenolic content in honey is a suitable factor to evaluate the quality and therapeutic potential of honey [74].

Also in this section, the pH of each honey sample was tested (Fig 7). The lowest pH was 1.9 and the highest pH was 5.2, and 48% of the total samples were in the pH range from 2.5 to 3.5. The global standard for the pH of honey is 3.9 (3.2 to 4.5). The pH values between 3.4 and 6.1 indicate the freshness of honey samples [75, 76]. Honey is recognized as a therapeutic option for the treatment of acute and chronic wounds, and a variety of in vitro and in vivo research studies have been performed in this area [77–79]. The low pH of honey (0.4 to 5.3) promotes wound healing [80, 81] which is accomplished by reducing the highly alkaline environment in non-healing wounds to a more acidic healing environment.

In this study, all honey samples showed a low moisture content (13% to 18.80%), which corresponded to the values ($< 20 <$) set by the Codex standard for honey (Alimentarius Codex, 1987). Moisture content is considered as an important criterion to evaluate the quality of honey. The less moisture in honey, the longer the shelf life [82]. The moisture content of honey depends on various factors such as the time of harvest, the climatic factors of the geographical area, and the storage temperature [83, 84]. The parameter of quality and uniqueness of honey has been studied by many researchers, and the quality and reliability of honey is very important to both buyers and producers.

Regression results between the average annual relative humidity of the area (independent variable) and the humidity of the honey sample (dependent variable) showed that there is a significant relationship between them (p-value 0.000), so the moisture content of the honey samples is a function of the environmental humidity (Fig 8).

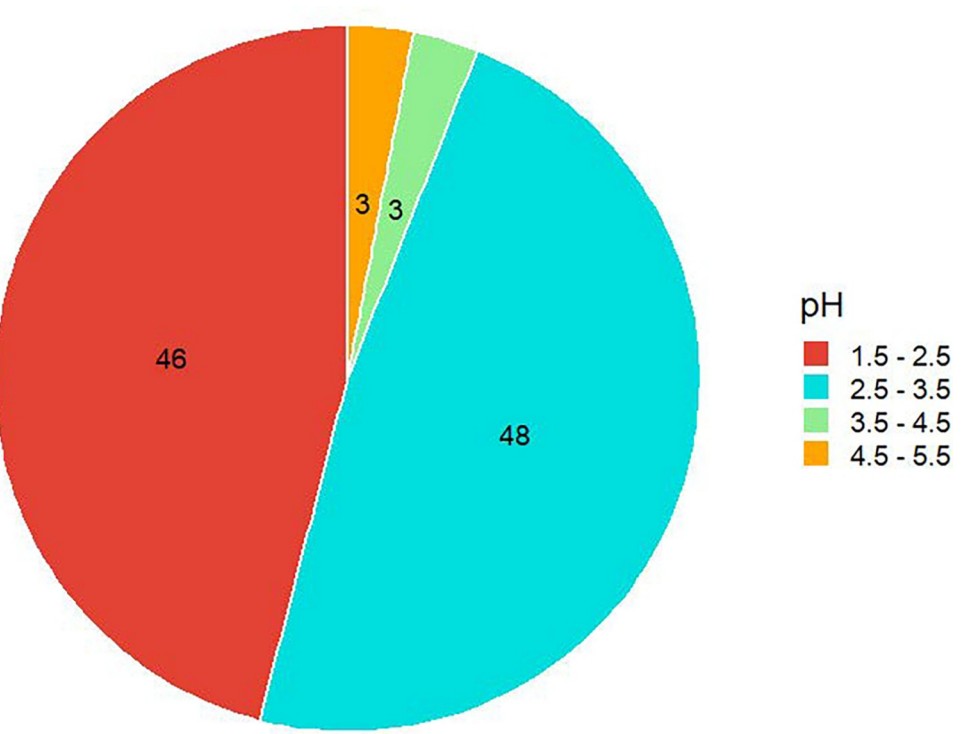

**Fig 7. pH abundance ratio of the samples.**

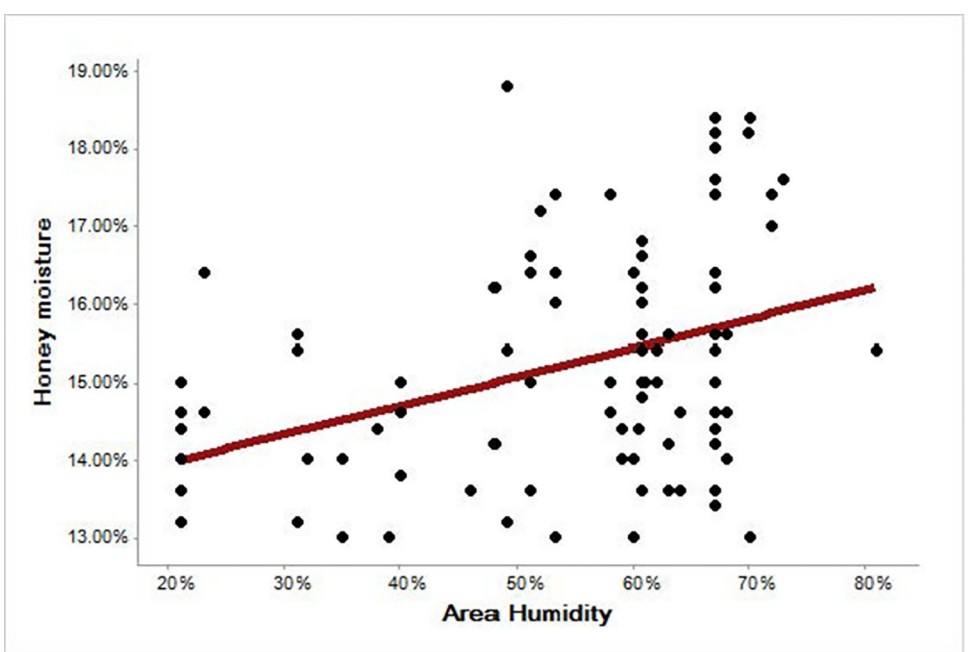

**Fig 8. Regression related to the effect of environmental humidity on the moisture of honey samples.**

**Table 2. Relative eigenvalue, percentage of variance, and vector values of principal component analysis (PCA) using total phenolic content, antioxidant activity, pH, and moisture content of the studied honey samples.**

| Name of variables | PC 1 | PC 2 | PC 3 | PC 4 |
|---|---|---|---|---|
| Eigenvalue | 1.19215 | 1.09555 | 0.932069 | 0.780231 |
| % variance | 29.804 | 27.389 | 23.302 | 19.506 |
| Antioxidant activities | -0.15449 | **0.79953** | -0.2348 | -0.53081 |
| Total phenol content | **0.59267** | 0.056052 | 0.69854 | -0.39705 |
| pH | **0.58965** | **0.47616** | -0.1825 | 0.62632 |
| Moisture content | -0.52648 | 0.36179 | 0.65085 | 0.41026 |

## Principal component analysis (PCA)

For the PCA analysis, both the first and second axes accounted for 57.19% of the total variance (Table 2). The first principal component explained 29.8% of the variance, the main chemical variables of which were: total phenol content and pH (Table 2). The second axis explained 27.39% of the variance and antioxidant activities and pH were the most significant variables (Table 2).

The PCA did not result in any significant differences between all honey samples studied (Fig 9). Considering that the honey samples studied were collected from different geographical areas, samples from the same area were not placed next to each other in the plot. This shows that honey samples from different locations in the same area have different chemical compositions.

## Cluster analysis

According to the cluster analysis of the plant family composition, the honey samples collected from a geographical location were located far from each other in the dendrogram and were not located in the same cluster, as shown in Fig 10.

One of the important concerns of honey quality is to guarantee the authenticity of honey according to legal requirements. This authenticity is achieved by recognizing the plant's origin, geography, and chemical analysis [85, 86] Nutritional and healing properties of honey from

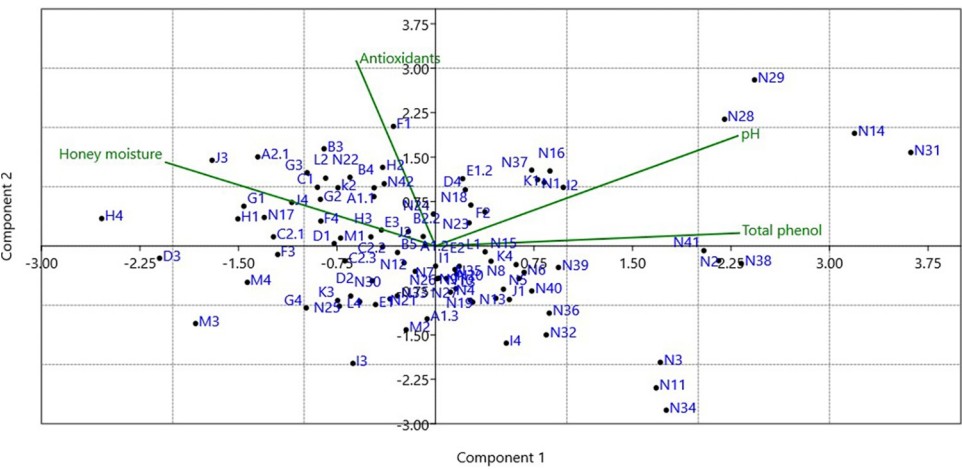

**Fig 9. Principal component analysis (PCA) scatter plot of the studied honey samples based on total phenolic content, antioxidant activity, pH, and moisture content.**

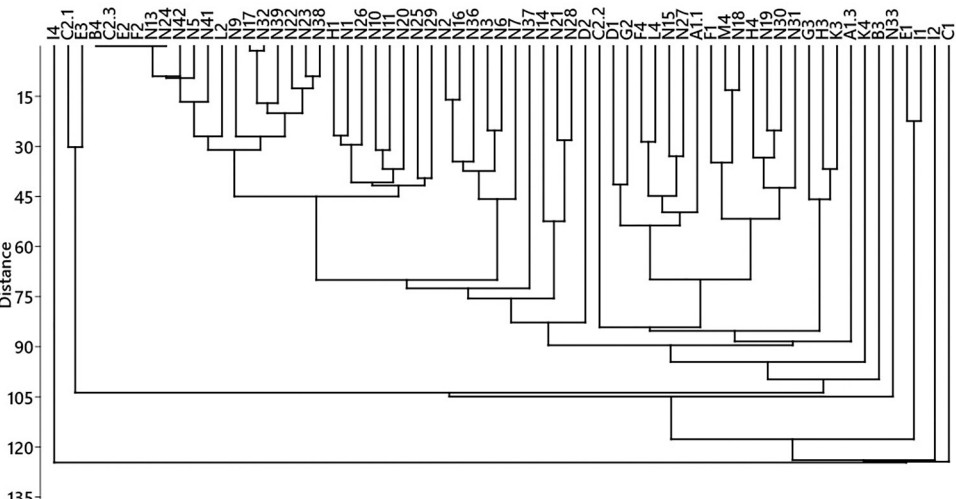

**Fig 10. Dendrogram of the studied honey samples according to their botanical family composition and clustered by UPGMA (unweighted pair group method using Euclidean distance).** Cophenetic correlation coefficient = 0.8565.

the very beginning make it an important component of traditional medicine [77, 87]. The natural antioxidant activities in honey are valuable against heart disease, cancer, inflammation, coronary heart disease, aging, tumors, gastrointestinal tract, and wound healing [88, 89]. In relation to health, honey, bee pollen, propolis, royal jelly, beeswax, and bee venom have all been used in traditional and modern medicine [90, 91]. Researchers have identified the bioactive properties of honey, propolis, and royal jelly, indicating compounds with antimicrobial, anti-inflammatory, antioxidant activity, anti-tumor and anti-cancer activities [90–92] Honey is used to treat stomach ulcers and injuries, increase oral health, fight stomach disorders, and liver and pancreatic diseases, as well as to promote cardiovascular health [90, 92].

## Conclusion

Melissopalynology is a useful and complementary tool along with chemical assays to measure the authenticity, fraud, and differentiation between different types of honey. In addition to being a useful tool in the food industry and therapeutic debates, melissopalynology is even more important in the conservation debate. Conducting more extensive studies in the field of melissopalynology in the future to identify the plant species used by bees is of mutual importance in the conservation debate, plant species conservation protects bees and bee conservation leads to plant species conservation. In the field of conservation, more and more ongoing research is needed to better understand the behavior and role of bees and plants. Bee populations are threatened by human activities. Climate change, invasive species, monocultures that have less variety in flower resources, and the use of pesticides have a negative effect on bees. Considering that climate changes affect the feeding behavior of mammals, these changes can affect insects more [93]. On the other hand, seasonal changes (including changes in temperature, natural light, and precipitation) and changes in vegetation type can not only have a significant impact on the behavior of bees but also on the behavior of other animals [94]. It is recommended that future studies look at the impact of climate change on honey bees and on the pollen of honey produced by farmed bees be compared to honey produced by wild bees to help better understand the role of melissopalynology in the protection of bees and plant species and the protection of ecosystems.

## Acknowledgments

The authors thank anonymous reviewers for their valuable comments on an earlier draft. We are grateful to Mr. Saeed Javadi Anaghizi is thanked for his assistance in SEM work. We are profoundly grateful to Mr. Mehdi Kia Hairati, Mr. Mohsen Yasari, Mr. Erfan Rajabi, and Mr. Mohammad Reza Kashfi for their valuable help during the fieldwork study.

## Statement

In the above study, the beekeepers provided us with honey samples voluntarily and with full consent, and we did not need to obtain a permit due to not entering protected areas or national parks.

## Author Contributions

**Conceptualization:** Ahmadreza Mehrabian, Dariush Minai, Farid Salmanpour.

**Data curation:** Zahra Shakoori.

**Formal analysis:** Zahra Shakoori.

**Investigation:** Zahra Shakoori, Dariush Minai, Farid Salmanpour.

**Methodology:** Zahra Shakoori, Ahmadreza Mehrabian, Dariush Minai.

**Project administration:** Zahra Shakoori, Farid Salmanpour.

**Software:** Zahra Shakoori, Farid Salmanpour, Farzaneh Khajoei Nasab.

**Supervision:** Zahra Shakoori, Ahmadreza Mehrabian, Farid Salmanpour.

**Writing – original draft:** Zahra Shakoori.

**Writing – review & editing:** Zahra Shakoori, Ahmadreza Mehrabian, Dariush Minai, Farid Salmanpour, Farzaneh Khajoei Nasab.

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
