## [Decision Letter · Decision Letter 0]

29 Mar 2023

PONE-D-22-31247Assessing the Quality of Bee Honey on the basis of melissopalynology as well chemical analysis: a geobotanic approachPLOS ONE

Dear Dr. Mehrabian,

Thank you for submitting your manuscript to PLOS ONE. After careful consideration, we feel that it has merit but does not fully meet PLOS ONE’s publication criteria as it currently stands. Therefore, we invite you to submit a revised version of the manuscript that addresses the points raised during the review process.

We look forward to receiving your revised manuscript.

Kind regards,

Daniel de Paiva Silva, Ph.D.

Academic Editor

PLOS ONE

Plant Biogeography and Vegetation of High Mountains of Central and South-West Asia - https://link.springer.com/book/10.1007/978-3-030-45212-4, and

A bioclimatic characterization of high elevation habitats in the Alborz mountains of Iran - https://doi.org/10.1007/s00035-018-0202-9.

In your revision ensure you cite all your sources (including your own works), and quote or rephrase any duplicated text outside the methods section. Further consideration is dependent on these concerns being addressed.

4. Please update your submission to use the PLOS LaTeX template. The template and more information on our requirements for LaTeX submissions can be found at http://journals.plos.org/plosone/s/latex.

6. PLOS requires an ORCID iD for the corresponding author in Editorial Manager on papers submitted after December 6th, 2016. Please ensure that you have an ORCID iD and that it is validated in Editorial Manager. To do this, go to ‘Update my Information’ (in the upper left-hand corner of the main menu), and click on the Fetch/Validate link next to the ORCID field. This will take you to the ORCID site and allow you to create a new iD or authenticate a pre-existing iD in Editorial Manager. Please see the following video for instructions on linking an ORCID iD to your Editorial Manager account: https://www.youtube.com/watch?v=_xcclfuvtxQ.

Additional Editor Comments:

Dear Dr. Mehrabian,

After this first review round, all three reviewers raised significant issues that prevent the publication of your manuscript at this time. Still, the reviewers indicated that the manuscript has a potential to be published in PLoS One after improvements are considered. Please resubmit your manuscript by June 29th, 2023 or in an earlier date in case you are able to conduct the changes faster than I have previously imagined. Please do not hesitate to contact me in case you need further time or have any other doubt that needs to be taken care of.

Sincerely,

Daniel Silva.

Reviewers' comments:

Reviewer's Responses to Questions

**Comments to the Author**

1. Is the manuscript technically sound, and do the data support the conclusions?

Reviewer #1: Partly

Reviewer #2: Yes

Reviewer #3: Yes

2. Has the statistical analysis been performed appropriately and rigorously? 

Reviewer #1: N/A

Reviewer #2: No

Reviewer #3: Yes

3. Have the authors made all data underlying the findings in their manuscript fully available?

Reviewer #1: Yes

Reviewer #2: Yes

Reviewer #3: Yes

4. Is the manuscript presented in an intelligible fashion and written in standard English?

Reviewer #1: No

Reviewer #2: No

Reviewer #3: No

5. Review Comments to the Author

Reviewer #1: Review of PONE-D-22-31247

In the article, melissopalynology and chemical analysis to evaluate 22 different honey samples from two main Iranian hubs of honey. Sampling was carried out at two 23 important poles of Iranian honey production, the central Alborz region, and the mountain ecosystems of Zagros in the years 2020-2021. results showed that all honey samples were classified as polyfloral honey. Based on their findings, 57 honey samples (61%) contained the standard amount of pollen. A total of 42 plant families and 55 genera were identified in the studied samples, with 32 the highest presence of Asteraceae, Fabaceae, Rosaceae, Apiaceae, and Apocynaceae.

To the article, I have next comments and recommendations:

• In the article, there are many typing errors, which should be carefully checked and corrected, e.g., in Abstract: you should use 0.08 to 0.05 ppm or 0.08-0.05 and so on throughout the text. Like wise L. 47:…climate of the region…; L. 61:…botanical…;, Table 1: Bignoniaceae L. 98: instead antioxidative should be used antioxidant activity or antioxidant capacity and similar in the all article.

• Latin names of plants should be in Italics through all text and References, e.g., L. 140: Quercus macranthera, etc.

• All applied equipment and chemicals should be carefully completed with the name of the producer, town and country (state) – check and complete the missing date especially in section Laboratory studies.

• In the centrifugation convert rpm to g values and correct centrifugation instead centrifuge(s), where it is appropriate.

• L. 637: 1% ferric chloride…

• Unify Apocynaceae & Apiaceae or Apocynaceae and Apiaceae, similarly throughout the text.

• L. 294 and elsewhere: total phenolic content or total phenolics.

• Figure 6: should be modified, the data on axis x are nearly not readable.

• Some sentences are completely confusing and should be corrected, e.g. L. 336-337:…in the Estonia (Puusepp and Koff 2014). Croatia (?). Our results show…

• The main problem is that the citation system should be substantially redone according to system used in PLOS One, i.e., using brackets like [1�, [2�,[3�, etc. and hence the references they will not be in alphabetical order.

• According to this system you should change the order of references. In References you should apply Italics in Latin names of plants (Prunus avium, Schefflera abyssinica) and bees (Apis mellifera), use the full titles of Journals with capitals of nouns and adjectives, like Food Chemistry, Nutrition Research, International Journal of Food Microbiology, Current Medicinal Chemistry, LWT-Food Science and Technology, etc., do not use Italics in titles of journals (not mixture). Do not use capitals in the name of article – should be e.g., Study of organoleptic, physical, and chemical indicators of honey in Tashkent region. Do not use pp. and p. in references, by the way some articles in new volumes use an article number (one number) instead of pages.

• Books, Theses and Website should be implemented along journals referenced with their citation number.

Here you are some examples:

1. Hailu D, Belay A. Melissopalynology and antioxidant properties used to differentiate Schefflera abyssinica and polyfloral honey. PLOS One. 2020;15(10): e0240868.https://doi.org/10.1371/journal.pone.0240868

2. Alvarez-Suarez MJ., Giampieri F, Battino M. Honey as a source of dietary antioxidants: structures, bioavailability and evidence of protective effects against human chronic diseases. Current Medicinal Chemistry. 2013; 20(5), 621-638. https://doi.org/10.2174/092986713804999358

3. Berberian M.The southern Caspian: a compressional depression floored by a trapped, modified oceanic crust. Canadian Journal of Earth Sciences. 1983; 20(2): 163-183. https://doi.org/10.1139/e83-015

4. Chua LS, Rahaman NL, Adnan NA, Eddie Tan, TT. Antioxidant activity of three honey samples in relation with their biochemical components. Journal of Analytical Methods in Chemistry. 2013: 313798. https://doi.org/10.1155/2013/313798

5. El-Senduny FF, Hegazi NM, Abd Elghani GE, Farag MA. Manuka honey, a unique mono-floral honey. A comprehensive review of its bioactives. metabolism, action mechanisms, and therapeutic merits. Food Bioscience. 2021; 42:101038. https://doi.org/10.1016/j.fbio.2021.101038

6. Hedge IC, Wendelbo P. Patterns of distribution and endemism in Iran. Notes from the Royal Botanic Garden, Edinburgh. 1978; 36(2): 441-464. https://www.scopus.com/record/display.uri?eid=2-s2.0-0006003564&origin=inward&txGid=08c1677fbb653bcda02974e2f9bd74ed

7. Kędzierska-Matysek M, Stryjecka M, Teter A, Skałecki P, Domaradzki P, Florek M. 2021. Relationships between the content of phenolic compounds and the antioxidant activity of Polish honey varieties as a tool for botanical discrimination. Molecules. 2021; 26(6): 1810. https://doi.org/10.3390/molecules26061810

Reviewer #2: Dear Authors,

The article is very interesting and seems to be balanced and well-structured.

However, I have some suggestions which can help you to improve the work, please find them below:

1. Authors stated in line 105 that “the study tries to analyze the mentioned data in relation to geographical factors.” and they try to analyze this aspect in the section 5. Discussion. However, for drawing conclusions if chemical parameters of honey depend on geographical factors, the proper statistical analysis is needed.

2. The sentence “We compared the humidity of honey samples with the humidity of the areas from which the samples were collected, which shows that the humidity of the environment affected the humidity of honey samples” must be proven by the statistical analysis.

3. On Figure. 6 the font size on the axis descriptions must be increased for greater readability. It is also no clear for me, what is the Loading. Do Authors mean coefficient of correlation?

4. The sentence in lines 299-302 should be corrected. It is completely incomprehensible.

5. Figure 8 – Autors should inform where the reader can find the explanation of codes of samples. On this picture Authors depict results of cluster analysis but I can’t find in the text how these results are used, or which conclusions were drawn base on them.

6. Text must be linguistically and editorially corrected.

Reviewer #3: The information contained in the article is excellent and the research sample is large and varied, but it needs to be rephrased in proper English

Notes attached

- Introduction

- The introduction consists of many staccato sentences, forcing the author to use an infinite number of linking phrases, such as: as well as , Furthermore, in addition, Therefore etc (Rewrite the entire introduction in proper English)

o Line 49 remove( the)

o Line 51 use counterfeit instead of fake

o Line 54 use and instead of as well as and change has to have

o Line 56 (Louveaux (1970, 1978)) Add reference

o Line 59- 65 Rewrite this paragraph (There are many separate sentences that must be linked together and rewritten in proper English)

o Line 73 (SW Asia) It was mentioned here for the first time, so it must be written without abbreviation

o Rewrite the paragraph from line 69-78

- Study area

- Rewrite in a proper English

- Field studies Change to sample collection

o Line 179 What is the meaning of amount of hight

- Laboratory studies

o It is better to put a paragraph for the chemicals and reagents used and another paragraph for the devices and tools used in the experiment separately

o Rewrite in a proper English

o Line 192 Then, remove large …. By passed through

- Line 223 DPPH abbreviation for what? It was mentioned here for the first time, so it must be written without abbreviation

- Line 237( 1% ) Repeated twice in the sentence why?

- Laboratory studies and chemical analysis of honey

o Scientifically rephrased

- Results

o Rewrite scientifically in a proper English

o Add a table containing honey samples names, symbols, and geographical areas

o Line 273 Was in the range instead of was the lowest

o Figure 6 Not clear

o Line 299-304 This part is incomprehensible and needs reformulation

- Unify the method of writing references within the article

- Line 348 Rephrasing

6. PLOS authors have the option to publish the peer review history of their article (what does this mean?). If published, this will include your full peer review and any attached files.

Reviewer #1: **Yes: **Prof. Ing. Jaromir Lachman, Ph.D.

Reviewer #2: No

Reviewer #3: **Yes: **Haya i. Aljohar

---

## [Author Response · Author response to Decision Letter 0]

25 May 2023

Dear Dr. Daniel de Paiva Silva

Thank you for providing us with the opportunity to submit a revised draft of the manuscript PONE-D-22-31247 entitled " Assessing the Quality of Bee Honey on the basis of melissopalynology as well chemical analysis ". We appreciate the time and effort that you and the reviewers have dedicated to providing valuable feedback on our manuscript. We are grateful to the reviewers for their insightful comments on the paper. We have been able to incorporate changes to reflect all the suggestions provided and all changes are highlighted in the manuscript text file. 

Please see below a point-by-point response to the reviewers’ comments and concerns. 

Best wishes 

Ahmadreza Mehrabian

Dear Editor 

Comments to the Author:

After this first review round, all three reviewers raised significant issues that prevent the publication of your manuscript at this time. Still, the reviewers indicated that the manuscript has a potential to be published in PLoS One after improvements are considered.

Response: We appreciate the positive comments of the reviewers and have done our best to incorporate all comments, with specific responses below. 

If applicable, we recommend that you deposit your laboratory protocols in protocols.io to enhance the reproducibility of your results. Protocols.io assigns your protocol its own identifier (DOI) so that it can be cited independently in the future.

Response: We saved our data in the figshare repository and placed the link in the text of the manuscript.

DATA AVAILABILITY STATEMENT

DOI: 10.6084/m9.figshare.22586863 

https://figshare.com/s/949e02f3163f39991d4e

Journal requirements:

Response: We prepared the manuscript with PLOS ONE style requirements, including file naming.

Plant Biogeography and Vegetation of High Mountains of Central and South-West Asia - https://link.springer.com/book/10.1007/978-3-030-45212-4, and

A bioclimatic characterization of high elevation habitats in the Alborz mountains of Iran - https://doi.org/10.1007/s00035-018-0202-9.

In your revision ensure you cite all your sources (including your own works), and quote or rephrase any duplicated text outside the methods section. Further consideration is dependent on these concerns being addressed.

Response: We added "A bioclimatic characterization of high elevation habitats in the Alborz mountains of Iran - https://doi.org/10.1007/s00035-018-0202-9" in our study area section:

The Alborz mountain range is located at 36° N from the Caspian Sea up to a height of 5671 meters (Noroozi et al.2018).

Response: In the above study, we did not need a permit because the honey samples were provided to us voluntarily by the beekeepers, and because there was no need to enter protected areas or national parks, we did not need to obtain a permit.

Statement: In the above study, the beekeepers provided us with honey samples voluntarily and with full consent, and we did not need to obtain a permit due to not entering protected areas or national parks. 

4. Please update your submission to use the PLOS LaTeX template. The template and more information on our requirements for LaTeX submissions can be found at http://journals.plos.org/plosone/s/latex.

Response: Corrections were made.

Response: We saved our laboratory protocols and data in the figshare repository and placed the link in the text of the manuscript.

DATA AVAILABILITY STATEMENT

DOI: 10.6084/m9.figshare.22586863 

https://figshare.com/s/949e02f3163f39991d4e

6. PLOS requires an ORCID iD for the corresponding author in Editorial Manager on papers submitted after December 6th, 2016. Please ensure that you have an ORCID iD and that it is validated in Editorial Manager. To do this, go to ‘Update my Information’ (in the upper left-hand corner of the main menu), and click on the Fetch/Validate link next to the ORCID field. This will take you to the ORCID site and allow you to create a new iD or authenticate a pre-existing iD in Editorial Manager. Please see the following video for instructions on linking an ORCID iD to your Editorial Manager account: https://www.youtube.com/watch?v=_xcclfuvtxQ.

Response: The ORCID of each author was declared.

Zahra Shakoori: https://orcid.org/0000-0003-3057-8981

Ahmadreza Mehrabian: https://orcid.org/0000-0001-6633-3092

Dariush Minai: https://orcid.org/0000-0003-3589-7324

Farid Salmanpour: https://orcid.org/0000-0002-1517-9061

Farzaneh Khajoei Nasab: https://orcid.org/0000-0002-2325-9555

Dear Prof. Ing. Jaromir Lachman, Reviewer #1:

To the article, I have next comments and recommendations:

• In the article, there are many typing errors, which should be carefully checked and corrected, e.g., in Abstract: you should use 0.08 to 0.05 ppm or 0.08-0.05 and so on throughout the text. Like wise L. 47:…climate of the region…; L. 61:…botanical…;, Table 1: Bignoniaceae L. 98: instead antioxidative should be used antioxidant activity or antioxidant capacity and similar in the all article.

Response: We tried to carefully check and correct typographical errors. For example, we used 0.08 to 0.05 pp throughout the text. Instead of antioxidant, we used antioxidant activity in the article. Other mentioned points such as amendment in L. 46:...climate of the region...; L. 58:…botanical…;, Table 1: Bignoniaceae done.

• Latin names of plants should be in Italics through all text and References, e.g., L. 140: Quercus macranthera, etc.

Response: The Latin names of the plants were written in italics in all texts and references.

• All applied equipment and chemicals should be carefully completed with the name of the producer, town and country (state) – check and complete the missing date especially in section Laboratory studies.

Response: The name of the manufacturer, city and country (state) of all equipment and chemicals used were carefully entered with the name of the manufacturer, especially in the laboratory studies section.

- L.212: Hot plate model HP100 made by Mtops company in South Korea

- L.216: Centrifuge model TDL-4 made by Jenius China

- L.247: Merck Chemical Company in Germany.

- L.248: 2,2-Diphenyl-1-picrylhydrazyl (DPPH) was obtained from Sigma American Chemical.

- 258: JENUS V-1100 spectrophotometer Made by DLAB China

• In the centrifugation convert rpm to g values and correct centrifugation instead centrifuge(s), where it is appropriate.

Response: In the centrifuge, we converted revolutions per minute to g values and corrected centrifugation instead of centrifuge where appropriate.

• L. 637: 1% ferric chloride…

Response: The item mentioned in L. 637 was not found, but it was found in line 264 and corrected.

• Unify Apocynaceae & Apiaceae or Apocynaceae and Apiaceae, similarly throughout the text.

Response: Apocynaceae and Apiaceae are similarly united throughout the text.

• L. 294 and elsewhere: total phenolic content or total phenolics.

Response: Throughout the text, the phenol compounds and total phenol were changed to total phenolic content.

• Some sentences are completely confusing and should be corrected, e.g. L. 336-337:…in the Estonia (Puusepp and Koff 2014). Croatia (?). Our results show…

Response: Confusing sentences were completely corrected throughout the text.

• The main problem is that the citation system should be substantially redone according to system used in PLOS One, i.e., using brackets like [1�, [2�,[3�, etc. and hence the references they will not be in alphabetical order.

Response: In this version, we have adjusted the entire manuscript according to PLOS One guidelines.

• According to this system you should change the order of references. In References you should apply Italics in Latin names of plants (Prunus avium, Schefflera abyssinica) and bees (Apis mellifera), use the full titles of Journals with capitals of nouns and adjectives, like Food Chemistry, Nutrition Research, International Journal of Food Microbiology, Current Medicinal Chemistry, LWT-Food Science and Technology, etc., do not use Italics in titles of journals (not mixture). Do not use capitals in the name of article – should be e.g., Study of organoleptic, physical, and chemical indicators of honey in Tashkent region. Do not use pp. and p. in references, by the way some articles in new volumes use an article number (one number) instead of pages.

Response: We modified and prepared all the references according to the instructions of the journal and all the mentioned points were applied.

• Books, Theses and Website should be implemented along journals referenced with their citation number.

Here you are some examples:

1. Hailu D, Belay A. Melissopalynology and antioxidant properties used to differentiate Schefflera abyssinica and polyfloral honey. PLOS One. 2020;15(10): e0240868.https://doi.org/10.1371/journal.pone.0240868

2. Alvarez-Suarez MJ., Giampieri F, Battino M. Honey as a source of dietary antioxidants: structures, bioavailability and evidence of protective effects against human chronic diseases. Current Medicinal Chemistry. 2013; 20(5), 621-638. https://doi.org/10.2174/092986713804999358

3. Berberian M.The southern Caspian: a compressional depression floored by a trapped, modified oceanic crust. Canadian Journal of Earth Sciences. 1983; 20(2): 163-183. https://doi.org/10.1139/e83-015

4. Chua LS, Rahaman NL, Adnan NA, Eddie Tan, TT. Antioxidant activity of three honey samples in relation with their biochemical components. Journal of Analytical Methods in Chemistry. 2013: 313798. https://doi.org/10.1155/2013/313798

5. El-Senduny FF, Hegazi NM, Abd Elghani GE, Farag MA. Manuka honey, a unique mono-floral honey. A comprehensive review of its bioactives. metabolism, action mechanisms, and therapeutic merits. Food Bioscience. 2021; 42:101038. https://doi.org/10.1016/j.fbio.2021.101038

6. Hedge IC, Wendelbo P. Patterns of distribution and endemism in Iran. Notes from the Royal Botanic Garden, Edinburgh. 1978; 36(2): 441-464. https://www.scopus.com/record/display.uri?eid=2-s2.0-0006003564&origin=inward&txGid=08c1677fbb653bcda02974e2f9bd74ed

7. Kędzierska-Matysek M, Stryjecka M, Teter A, Skałecki P, Domaradzki P, Florek M. 2021. Relationships between the content of phenolic compounds and the antioxidant activity of Polish honey varieties as a tool for botanical discrimination. Molecules. 2021; 26(6): 1810. https://doi.org/10.3390/molecules26061

Response: We have made the mentioned corrections completely.

Dear Reviewer #2:

1. Authors stated in line 105 that “the study tries to analyze the mentioned data in relation to geographical factors.” and they try to analyze this aspect in the section 5. Discussion. However, for drawing conclusions if chemical parameters of honey depend on geographical factors, the proper statistical analysis is needed.

Response: This section has been completely revised.

2. The sentence “We compared the humidity of honey samples with the humidity of the areas from which the samples were collected, which shows that the humidity of the environment affected the humidity of honey samples” must be proven by the statistical analysis.

Response: We removed the map related to the effect of ambient humidity on the humidity of honey samples and using the minitab software, we got regression fitted line plots statistical model.

The regression result between the average annual relative humidity of the region (independent variable) and the humidity of the honey sample (dependent variable) showed that there is a significant relationship (p value 0.000) between them, so that the humidity of honey samples is a function of the humidity of the environment.

Fig 5. Comparison of moisture content of honey samples with moisture content of the collection site

Fig 5. Regression related to the effect of environmental humidity on the moisture of honey samples

3. On Figure. 6 the font size on the axis descriptions must be increased for greater readability. It is also no clear for me, what is the Loading. Do Authors mean coefficient of correlation?

Response: In this section, more detailed reforms were needed, and these reforms were carried out.

Yes, loading means the same coefficient of correlation that was modified in the figure.

4. The sentence in lines 299-302 should be corrected. It is completely incomprehensible.

Response: In this section, more detailed reforms were needed, and these reforms were carried out.

5. Figure 8 – Authors should inform where the reader can find the explanation of codes of samples. On this picture Authors depict results of cluster analysis but I can’t find in the text how these results are used, or which conclusions were drawn base on them.

Response: We saved our data in the figshare repository and placed the link in the text of the manuscript, codes can be found in it.

DOI: 10.6084/m9.figshare.22586863 

https://figshare.com/s/949e02f3163f39991d4e

In addition, this section was modified in the text of the manuscript and some information was included in the text of the manuscript.

6. Text must be linguistically and editorially corrected.

Response: The text was linguistically and editorially corrected.

Dear Dr. Haya i. Aljohar, Reviewer #3

- Introduction

- The introduction consists of many staccato sentences, forcing the author to use an infinite number of linking phrases, such as: as well as , Furthermore, in addition, Therefore etc (Rewrite the entire introduction in proper English).

Response: We rewrote the entire introduction in proper English.

o Line 49 remove( the)

Response: It was corrected.

o Line 51 use counterfeit instead of fake

Response: Counterfeit replaced fake.

o Line 54 use and instead of as well as and change has to have.

Response: It was corrected

o Line 56 (Louveaux (1970, 1978)) Add reference

Response: Line59: Reference added (Louveaux et al. 1970; 1978)

o Line 59- 65 Rewrite this paragraph (There are many separate sentences that must be linked together and rewritten in proper English).

Response: We have rewritten this paragraph in proper English.

o Line 73 (SW Asia) It was mentioned here for the first time, so it must be written without abbreviation.

Response: Line 78: Southwest Asia is written instead of SW Asia

o Rewrite the paragraph from line 69-78.

Response: The text of the manuscript was completely rewritten and revised

- Study area

- Rewrite in a proper English

Response: All the text of the manuscript was rewritten in proper English.

- Field studies Change to sample collection

Response: Field studies Changed to sample collection

o Line 179 What is the meaning of amount of hight.

Response: We mean the same height. 

Line 197: The amount of hight was changed to height

- Laboratory studies

o It is better to put a paragraph for the chemicals and reagents used and another paragraph for the devices and tools used in the experiment separately.

Response: We have completely revised this paragraph and tried to separate the content as much as possible in the same paragraph to make it clearer.

o Rewrite in a proper English.

Response: All the text of the manuscript was rewritten in proper English.

o Line 192 Then, remove large …. By passed through

Response: corrections were made

o Line 223 DPPH abbreviation for what? It was mentioned here for the first time, so it must be written without abbreviation.

Response: Line 248: DPPH was written in full.

2,2-Diphenyl-1-picrylhydrazyl (DPPH) was obtained from Sigma American Chemical Company.

- Line 237( 1% ) Repeated twice in the sentence why?

Response: FeCl3 1% is the abbreviation of 1% ferric chloride solution, that's why 1% is repeated twice in the sentence.

Line 264: The sentence was modified in this way: In this experiment, 1% ferric chloride solution (FeCl3 1%)...

- Results

o Rewrite scientifically in a proper English

Response: It was scientifically rewritten in appropriate English

o Add a table containing honey samples names, symbols, and geographical areas

Response: We have added the requested information and stored it in the figshare repository as backup information.

DATA AVAILABILITY STATEMENT

DOI: 10.6084/m9.figshare.22586863 

https://figshare.com/s/949e02f3163f39991d4e

o Line 273 Was in the range instead of was the lowest.

Response: Line 339: It was corrected.

o Figure 6 Not clear

Response: Response: In this section, more detailed reforms were needed, and these reforms were carried out.

o Line 299-304 This part is incomprehensible and needs reformulation

Response: Line 424 – 427: The paragraph has been completely revised.

The PCA did not result in any significant differences between all honey samples studied (Fig. 9). Considering that the honey samples studied were collected from different geographical areas, samples from the same area were not placed next to each other in the plot. This shows that honey samples from different locations in the same area have different chemical compositions.

- Unify the method of writing references within the article

Response: Unify the references in the article were prepared as based on the journal's guidelines.

- Line 348 Rephrasing

Response: Rewriting done

---

## [Decision Letter · Decision Letter 1]

26 Jun 2023

PONE-D-22-31247R1Assessing the Quality of Bee Honey on the basis of melissopalynology as well chemical analysisPLOS ONE

Dear Dr. Mehrabian,

Thank you for submitting your manuscript to PLOS ONE. After careful consideration, we feel that it has merit but does not fully meet PLOS ONE’s publication criteria as it currently stands. Therefore, we invite you to submit a revised version of the manuscript that addresses the points raised during the review process.

We look forward to receiving your revised manuscript.

Kind regards,

Daniel de Paiva Silva, Ph.D.

Academic Editor

PLOS ONE

Additional Editor Comments:

Dear Dr. Mehrabian,

After this new review round, one of the reviewers still believes your manuscripts deserves a major review. Among several issues, the reviewer believes (and I agree) that you manuscript needs an in-deep review and proofreading. Therefore, considering the issues raised by this reviewer (Reviewer #3), I will grant you a two-month period to deliver the new review for the appreciation of the reviewer.

Sincerely,

Daniel Silva

Reviewers' comments:

Reviewer's Responses to Questions

**Comments to the Author**

1. If the authors have adequately addressed your comments raised in a previous round of review and you feel that this manuscript is now acceptable for publication, you may indicate that here to bypass the “Comments to the Author” section, enter your conflict of interest statement in the “Confidential to Editor” section, and submit your "Accept" recommendation.

Reviewer #1: All comments have been addressed

Reviewer #2: All comments have been addressed

Reviewer #3: (No Response)

2. Is the manuscript technically sound, and do the data support the conclusions?

Reviewer #1: Partly

Reviewer #2: Yes

Reviewer #3: Yes

3. Has the statistical analysis been performed appropriately and rigorously? 

Reviewer #1: N/A

Reviewer #2: Yes

Reviewer #3: Yes

4. Have the authors made all data underlying the findings in their manuscript fully available?

Reviewer #1: Yes

Reviewer #2: Yes

Reviewer #3: No

5. Is the manuscript presented in an intelligible fashion and written in standard English?

Reviewer #1: Yes

Reviewer #2: Yes

Reviewer #3: No

6. Review Comments to the Author

Reviewer #1: Review of PONE-D-22-31247R1

The article has been in part improved; however I still have some little and larger reminders.

• L. 63:..dependent on…

• In the article you used both units: mL and uL or ml and ul. Please unify to one expression of the liter unit – l or L. I prefer mL and uL throughout the article, correct it.

• Table 1: Acanthaceae should be in Bold in the column Family.

• L. 263:..(Fig. 6).

• L. 290:..(Alimentarius Codex)…

• My main comments are about the citations and references, which need to be corrected. Citations of number in brackets: they should be from the start of the article text, i.e., from 1 to 95 and subsequently renumbered in References 1-95. So, References will not be arranged alphabetically, but from 1 to 95; so [1) will be Zimmermann MI ….; [2) Kečkeš S, Gašić U, Veličković TĆ, Milojković-Opsenica D, Natić M, Tešić Ž….; [3) Mehrabian A, Rad MA….; [4) Alvarez-Suarez JM…; [5) Louveaux J, Maurizio A, Vorwohl G…; [6) Louveaux J, Maurizio A, Vorwohl G. Methods of melissopalynology, etc. This way you can continue editing further.

• Another important item is the title of referenced journals. You can choose two different ways - either the full title of the journal or its abbreviation. However, the style should be unique and not a mixture.

• Here you are some examples: Food Chemistry (full) or Food Chem. (abbreviation); Nutrition Research (full) or Nutr Res. (abbreviation); International Journal of Food Microbiology (full) or Int J Food Microbiol. (abbreviation); Current Medicinal Chemistry (full) or Curr Med Chem. (abbreviation); Acta Botanica Fennica (full) or Acta Bot Fenn. (abbreviation); Veterinary Microbiology (full) or Vet Microbiol. (abbreviation). You should follow these corrections in References and choose only one method – either the full title or its abbreviation.

Reviewer #2: I have no comments because the authors have taken all my suggestions into account and have revised the manuscript in line with my expectations

Reviewer #3: ◦ The article need proofreading, especially the shaded parts

◦ The numbering of reference uncorrected

◦ Put all instruments and equipment under instrumentations.

◦ Put all chemicals and reagent under chemical and reagents.

◦ Delete everything marked as deleted.

◦ Line 21: Melissopalynological and chemical analysis of honey provide us useful

◦ Line 57: Comma instead of fullstop

◦ Line 113: fig. 1 Write complete caption, it must identify what’s the red spots.

◦ Line 17: in the period from

◦ Line 122: One sample

◦ Line 123: Remove comma.

◦ Line 123-124: Delete this part.

◦ Line 126: Start with - material or chemical and reagents - instrumentations.

◦ Line132-133: Then, the honey samples were filtrate.

◦ Line 135-140: Rewrite this paragraph. What is the speed and how many round per minutes?

◦ Line 151: from instead of by

◦ Line 156: Start the statement with For pollen identification we used …..

◦ Line 161: Put it under chemical and reagents.

◦ Line 171: add Place

◦ Line 177-180: 0.1 ml of ferric acid (FeCl3) was mixed with 0.1 ml of diluted honey and the absorbance was measured at 590 nm., against Phenol solution (0.09%) which was used as standard. []

◦ Line 185: Add information about this instrument under instrumentation.

◦ Line 190-193: proofreading

◦ Fig.3 alb, c,: Is it actual figure for your sample.

◦ Table 1: Add column for samples name or number.

7. PLOS authors have the option to publish the peer review history of their article (what does this mean?). If published, this will include your full peer review and any attached files.

Reviewer #1: **Yes: **Jaromir Lachman

Reviewer #2: No

Reviewer #3: **Yes: **Haya i. Aljohar

---

## [Author Response · Author response to Decision Letter 1]

2 Jul 2023

Dear Dr. Daniel de Paiva Silva

Thank you for providing us with the opportunity to submit a revised draft of the manuscript PONE-D-22-31247 entitled " Assessing the Quality of Bee Honey on the basis of melissopalynology as well chemical analysis ". We appreciate the time and effort that you and the reviewers have dedicated to providing valuable feedback on our manuscript. We are grateful to the reviewers for their insightful comments on the paper. We have been able to incorporate changes to reflect all the suggestions provided and all changes are highlighted in the manuscript text file. 

Please see below a point-by-point response to the reviewers’ comments and concerns. 

Best wishes 

Ahmadreza Mehrabian

Dear Editor 

Comments to the Author:

After this new review round, one of the reviewers still believes your manuscripts deserves a major review. Among several issues, the reviewer believes (and I agree) that you manuscript needs an in-deep review and proofreading. Therefore, considering the issues raised by this reviewer (Reviewer #3), I will grant you a two-month period to deliver the new review for the appreciation of the reviewer.

Response: We appreciate the positive comments of the reviewers and have done our best to incorporate all comments, with specific responses below. 

If applicable, we recommend that you deposit your laboratory protocols in protocols.io to enhance the reproducibility of your results. Protocols.io assigns your protocol its own identifier (DOI) so that it can be cited independently in the future.

Response: We saved our data in the figshare repository and placed the link in the text of the manuscript.

DATA AVAILABILITY STATEMENT

DOI: 10.6084/m9.figshare.22586863 

https://figshare.com/s/949e02f3163f39991d4e

Dear Prof. Ing. Jaromir Lachman, Reviewer #1:

• L. 63: dependent on…

Response: L. 63 it was corrected.

• In the article you used both units: mL and uL or ml and ul. Please unify to one expression of the liter unit – l or L. I prefer mL and uL throughout the article, correct it.

Response: All µl were changed to µL, and ml to mL according to the reviewer's opinion.

• Table 1: Acanthaceae should be in Bold in the column Family.

Response: It was corrected.

• L. 263: (Fig. 6).

Response: It was corrected.

• L. 290:(Alimentarius Codex)…

Response: L.294: It was corrected.

• My main comments are about the citations and references, which need to be corrected. Citations of number in brackets: they should be from the start of the article text, i.e., from 1 to 95 and subsequently renumbered in References 1-95. So, References will not be arranged alphabetically, but from 1 to 95; so [1) will be Zimmermann MI ….; [2) Kečkeš S, Gašić U, Veličković TĆ, Milojković-Opsenica D, Natić M, Tešić Ž….; [3) Mehrabian A, Rad MA….; [4) Alvarez-Suarez JM…; [5) Louveaux J, Maurizio A, Vorwohl G…; [6) Louveaux J, Maurizio A, Vorwohl G. Methods of melissopalynology, etc. This way you can continue editing further.

Response: All references were edited according to the opinion of the respected referee.

• Another important item is the title of referenced journals. You can choose two different ways - either the full title of the journal or its abbreviation. However, the style should be unique and not a mixture.

Here you are some examples: Food Chemistry (full) or Food Chem. (abbreviation); Nutrition Research (full) or Nutr Res. (abbreviation); International Journal of Food Microbiology (full) or Int J Food Microbiol. (abbreviation); Current Medicinal Chemistry (full) or Curr Med Chem. (abbreviation); Acta Botanica Fennica (full) or Acta Bot Fenn. (abbreviation); Veterinary Microbiology (full) or Vet Microbiol. (abbreviation). You should follow these corrections in References and choose only one method – either the full title or its abbreviation.

Response: It was corrected.

Dear Reviewer #2:

Thank you for your constructive and valuable comments and we are glad that we were able to implement your comments correctly and completely.

Dear Dr. Haya i. Aljohar, Reviewer #3:

• The numbering of reference uncorrected

Response: It was corrected.

• Put all instruments and equipment under instrumentations.

Response: L. 135-142 we put all the tools and equipment under the instrumentations.

• Put all chemicals and reagent under chemical and reagents.

Response: L.131-134 we placed all chemicals and reagents under chemicals and reagents.

• Delete everything marked as deleted.

Response: We removed everything marked as removed.

• Line 21: Melissopalynological and chemical analysis of honey provide us useful.

Response: It was corrected.

• Line 57: Comma instead of fullstop.

Response: It was corrected.

• Line 113: fig. 1 Write complete caption, it must identify what’s the red spots.

Response: L.117 All the sampling locations are marked using red dots.

In the legend of Figure 1, it is also explained about the red dots that indicate the sampling points.

• Line 117: in the period from

Response: L.121 it was corrected.

• Line 122: One sample

Response: L.126 it was corrected.

• Line 123: Remove comma.

Response: L.127 the comma was removed

• Line 123-124: Delete this part.

Response: L.127 – 128 this part was deleted

• Line 126: Start with - material or chemical and reagents - instrumentations.

Response: L.131 it was corrected.

Material or chemical and reagents

• Line132-133: Then, the honey samples were filtrate.

Response: L.152 it was corrected.

• Line 135-140: Rewrite this paragraph. What is the speed and how many round per minutes?

Response: L.154-159 it was corrected.

• Line 151: from instead of by

Response: L.171 From was used instead of by

• Line 156: Start the statement with For pollen identification we used …..

Response: L.176 it was corrected.

• Line 161: Put it under chemical and reagents.

Response: L.132 – 134 we put the sentence under chemicals and reagents

• Line 171: add Place

Response: It was corrected.

• Line 177-180: 0.1 ml of ferric acid (FeCl3) was mixed with 0.1 ml of diluted honey and the absorbance was measured at 590 nm., against Phenol solution (0.09%) which was used as standard. []

Response: L.197 – 198 it was corrected.

• Line 185: Add information about this instrument under instrumentation.

Response: Information about the refractometer was added in the instrumentation section.

• Line 190-193: proofreading

Response: L. 205-210 it was corrected.

Principal component analysis (PCA) was used to determine the relationship between the geographical origin of honey samples and the amount of chemical compounds they contained. The main objective of this analysis was to determine whether honey samples in a geographic area contained the same amount of chemical compounds. Chemical parameters (pH), chemical compounds (total phenolic content and antioxidant activity) and moisture content were used for PCA analysis. A variance-covariance matrix was generated for all honey samples and four chemical parameters. Data were analyzed using PAST software version 3.21[60].

• Fig.3 alb, c,: Is it actual figure for your sample.

Response: Yes, all the photos in Figure 3(a,b,c,d) are related to the pollen extracted from the honey samples in this study, which were imaged using a scanning electron microscope.

Table 1: Add column for samples name or number.

Response: In Table 1, a column was added for the number of samples.

• Is it actual figure for your sample if it is not please use the actual figure?

Response: Yes, all the Figures are related to the pollen extracted from the honey samples in this study, which were imaged using a scanning electron microscope.

---

## [Decision Letter · Decision Letter 2]

21 Jul 2023

PONE-D-22-31247R2Assessing the Quality of Bee Honey on the Basis of Melissopalynology as well as Chemical AnalysisPLOS ONE

Dear Dr. Mehrabian,

Thank you for submitting your manuscript to PLOS ONE. After careful consideration, we feel that it has merit but does not fully meet PLOS ONE’s publication criteria as it currently stands. Therefore, we invite you to submit a revised version of the manuscript that addresses the points raised during the review process.

We look forward to receiving your revised manuscript.

Kind regards,

Daniel de Paiva Silva, Ph.D.

Academic Editor

PLOS ONE

Journal Requirements:

Additional Editor Comments:

Dear Dr. Mehrabian,

In this second review round, the reviewer believes the MS should be accepted for publication in PLoS One after a minor review.

I agree with the provided criticisms but believe there are only minor improvements to be made before acceptance.

Congratulations on your hard work!

Sincerely,

Daniel Silva, PhD.

Reviewers' comments:

Reviewer's Responses to Questions

**Comments to the Author**

1. If the authors have adequately addressed your comments raised in a previous round of review and you feel that this manuscript is now acceptable for publication, you may indicate that here to bypass the “Comments to the Author” section, enter your conflict of interest statement in the “Confidential to Editor” section, and submit your "Accept" recommendation.

Reviewer #1: (No Response)

2. Is the manuscript technically sound, and do the data support the conclusions?

Reviewer #1: Yes

3. Has the statistical analysis been performed appropriately and rigorously? 

Reviewer #1: Yes

4. Have the authors made all data underlying the findings in their manuscript fully available?

Reviewer #1: Yes

5. Is the manuscript presented in an intelligible fashion and written in standard English?

Reviewer #1: Yes

6. Review Comments to the Author

Reviewer #1: Review of PONE-D-22-31247_R2

The article has been substantially improved according to reviewers´ remarks and recommendations. To the article, I have some remarks, regarding journal titles and names of articles in References, which can be easily corrected and then the article can be ready for acceptance:

• L. 422: Journal of Apicultural Research; L. 441-442: L. 445: Veterinary Microbiology; L. 450-452: Study of organoleptic, physical and chemical indicators of honey in Tashkent region. L. 551: Current Medicinal Chemistry; L. 554: Nutrition Research; L. 570: Alpine Botany; International Journal of Food Microbiology; L. 593: Journal of Analytical Methods in Chemistry; L. 598: Palaeontologia Electronica; L. 601: Comprehensive Reviews in Food Science and Food Safety; L. 613: Bee world; L. 626, 632, 655, 672, 675: Food Chemistry; L. 638: Food Control; L. 644-645: Journal of Food Science and Technology; L. 651: Journal of Applied Microbiology; L. 657-658: British Journal of Surgery; L. 698-699: Oxidative Medicine and Cellular Longevity.

7. PLOS authors have the option to publish the peer review history of their article (what does this mean?). If published, this will include your full peer review and any attached files.

Reviewer #1: **Yes: **Jaromír Lachman

---

## [Author Response · Author response to Decision Letter 2]

23 Jul 2023

Dear Dr. Daniel de Paiva Silva

Thank you for providing us with the opportunity to submit a revised draft of the manuscript PONE-D-22-31247 entitled "Assessing the Quality of Bee Honey on the basis of melissopalynology as well chemical analysis ". We appreciate the time and effort that you and the reviewers have dedicated to providing valuable feedback on our manuscript. We are grateful to the reviewer for his insightful comments on the paper. We have been able to incorporate changes to reflect all the suggestions provided and all changes are highlighted in the manuscript text file. 

Please see below a point-by-point response to the reviewers’ comments and concerns. 

Best wishes 

Ahmadreza Mehrabian

Dear Prof. Ing. Jaromir Lachman, Reviewer #1:

The article has been substantially improved according to reviewers´ remarks and recommendations. To the article, I have some remarks, regarding journal titles and names of articles in References, which can be easily corrected and then the article can be ready for acceptance:

• L. 422: Journal of Apicultural Research; L. 441-442: L. 445: Veterinary Microbiology; L. 450-452: Study of organoleptic, physical and chemical indicators of honey in Tashkent region. L. 551: Current Medicinal Chemistry; L. 554: Nutrition Research; L. 570: Alpine Botany; International Journal of Food Microbiology; L. 593: Journal of Analytical Methods in Chemistry; L. 598: Palaeontologia Electronica; L. 601: Comprehensive Reviews in Food Science and Food Safety; L. 613: Bee world; L. 626, 632, 655, 672, 675: Food Chemistry; L. 638: Food Control; L. 644-645: Journal of Food Science and Technology; L. 651: Journal of Applied Microbiology; L. 657-658: British Journal of Surgery; L. 698-699: Oxidative Medicine and Cellular Longevity.

Response: All the references were corrected and according to the opinion of the respected referee, the first letter of the names of the journals in the reference was also capitaled.

---

## [Editor Report · Decision Letter 3]

25 Jul 2023

Assessing the Quality of Bee Honey on the Basis of Melissopalynology as well as Chemical Analysis

PONE-D-22-31247R3

Dear Dr. Mehrabian,

We’re pleased to inform you that your manuscript has been judged scientifically suitable for publication and will be formally accepted for publication once it meets all outstanding technical requirements.

Kind regards,

Daniel de Paiva Silva, Ph.D.

Academic Editor

PLOS ONE
---

## [Editor Report · Acceptance letter]

28 Jul 2023

PONE-D-22-31247R3 

Assessing the Quality of Bee Honey on the Basis of Melissopalynology as well as Chemical Analysis 

Dear Dr. Mehrabian:

I'm pleased to inform you that your manuscript has been deemed suitable for publication in PLOS ONE. Congratulations! Your manuscript is now with our production department. 

Kind regards, 

on behalf of

Dr. Daniel de Paiva Silva 

Academic Editor

PLOS ONE